# Age Diversity in Neighborhoods—A Mixed-Methods Approach Examining Older Residents and Community Wellbeing

**DOI:** 10.3390/ijerph20166574

**Published:** 2023-08-13

**Authors:** Hanna Varjakoski, Samuli Koponen, Antti Kouvo, Elisa Tiilikainen

**Affiliations:** Faculty of Social Sciences and Business Studies, Department of Social Sciences, Kuopio Campus, University of Eastern Finland, FI-70211 Kuopio, Finland; samuli.koponen@uef.fi (S.K.); antti.kouvo@uef.fi (A.K.); elisa.tiilikainen@uef.fi (E.T.)

**Keywords:** ageing, age diversity, neighbourhoods, mixed methods, perceived neighbourhood satisfaction

## Abstract

This paper focuses on age diversity in neighbourhoods and its possible impacts on community wellbeing. The aims of this paper are (a) to investigate whether age diversity in neighbourhoods contribute to older residents’ wellbeing and (b) to explore older residents’ experiences and views on age diversity in their neighbourhood. These questions are addressed using a mixed-method approach combining survey and interview data and analysis. The data is derived from a survey (n = 420) and 19 semistructured interviews addressed to the older residents of a rental house company located in Eastern Finland. The interview data is analysed using qualitative content analysis. The results of qualitative data indicate that older adults see various benefits in an age-diverse living environment. In the quantitative analysis, we apply multilevel models in our statistical analyses to take both community- and individual-level variation into account. The quantitative results show that older adults living in age-diverse neighbourhoods reported higher community wellbeing. Such association was not found among the younger residents. Overall, our study adds to the understanding of the importance of neighbouring relations on community wellbeing in later life. The results can be utilized when developing age-friendly environments and housing policies at local and national levels.

## 1. Introduction

There are many studies that investigate the effects of social diversity in the neighbourhood, ethnic and economic diversity in particular. However, age diversity has received less attention although ‘age differences are the most natural dimension of neighbourhood diversity’ [1] (p. 819). Consequently, not much is known about the interplay between age diversity and community wellbeing in a context where intergenerational encounters occur naturally, such as in one’s immediate living environment. This article aims to shed some light on the topic from the perspective of older residents.

Due to demographic change and current ageing-in-place policies [2], a growing number of older people live in their private homes and local communities and reach advanced age. Moreover, in Western societies, an increasing number of older adults also live in urban neighbourhoods [3], alone (women aged 65+ in particular), and as renters [4,5].

Place of residence is often considered particularly significant for older people for several reasons. Older adults tend to spend more time at home and in the neighbourhood, and have often lived longer in the same locality, resulting in greater attachment to the local neighbourhood. With older age, individuals also often become more dependent on supportive neighbourhood relationships [6]. Consequently, there has been a substantial amount of interest in exploring aspects of physical and social neighbourhoods for community-dwelling older people and finding ways in which neighbourhoods can provide support for their older residents.

At least in Finland, less attention has been given to how the experiences of the neighbourhood between older and younger residents differ. This study focuses on age diversity in neighbourhoods and its possible impacts on community wellbeing. The aim of the study is to examine (a) whether age-diversity in neighbourhoods contribute to older residents’ community wellbeing and (b) to explore older residents’ experiences and views on age diversity in their neighbourhood. These questions are addressed using a mixed-method approach combining survey and interview data from the residents of a rental house company in a city in Eastern Finland. We start by outlining the background of the study, followed by a description of the methodological approach. After this, we present the results and conclude with suggestions for future research.

### 1.1. Ageing and Neighbouring

Existing studies indicate that neighbour relations often become more meaningful in later life and that neighbours can form a notable part of older people’s social landscape. Older people often turn to their neighbours when looking for informal personal relationships [7] (p. 555). Further, in comparison to younger age groups, retired people have more friends in the neighbourhood and they also have the broadest chatting network in their place of residence [8]. In some cases, neighbours and friends may even play a more significant role in supporting the everyday lives of older people than family members do [9] (p. 1448). Overall, good neighbours are seen as a source of joy, company, sense of security, and informal help [10,11] and, for many older adults, neighbours play a significant role in their wellbeing [7,12,13,14].

The neighbourhood is an important context for ageing and everyday life [13]. Studies show that older people can be especially attached to their neighbourhood [15] and that local neighbourhood characteristics are more significant for older people [16]. Some scholars have even suggested that neighbourhood is a major contributor to quality of life for older adults and that neighbourhood characteristics can be stronger predictors of residential satisfaction than housing features [17] (p. 439). Either way, the immediate environment often becomes more important as an individual ages [18] (p. 3). For instance, a British study has identified that living in a neighbourly and supportive area with good facilities to promote relationships with neighbours is among the main cornerstones of quality of life in later years [19]. Earlier studies have also identified the importance of neighbourhood characteristics to the health and wellbeing of older people who usually spend large proportions of their time in their neighbourhood and at home [20,21]. With ageing, the neighbourhood and its characteristics are likely to become even more central in providing sources to maintain social life and inclusion, health, wellbeing, and identity [18,20]. Thus, considering the importance of the neighbourhood to many older people, neighbourhood satisfaction can be a notable element in contributing to or reducing one’s wellbeing.

### 1.2. Neighbourhood Satisfaction and Community

One mechanism through which objective characteristics of neighbourhood can affect individuals’ subjective wellbeing is individuals’ evaluation or attitude about their neighbourhood, i.e., satisfaction with the neighbourhood [22,23]. In general, satisfaction can be considered in a holistic or partial manner. The holistic approach considers the neighbourhood in its completeness, while in the partial approach, the neighbourhood is considered based on separate aspects, such as noise, other residents, or aesthetics. In this article, we refer to this satisfaction as ‘community wellbeing’, which can be defined as ‘the satisfaction with the local place of residence taking into account the attachment to it, the social and physical environment, and the services and facilities’ [24] (p. 734).

As an environment, neighbourhoods consist of physical, social, and cultural properties [25]. Combining these properties with human psychology (e.g., choices and biases) renders neighbourhood satisfaction arguably a complex construct. This complexity allows debate on the determinants of neighbourhood satisfaction. For example, Barnett et al. argue that neighbourhood satisfaction is shaped by both physical (objective) and perceived (subjective) neighbourhood characteristics [26]. Examples of objective characteristics are crime and noise, which increased the odds of being dissatisfied with one’s neighbourhood [27]. Contrary to these findings, Lee and colleagues observed that neighbourhood satisfaction is driven mostly by perception, not objective reality [28]. This perception “carries more weight” when explaining neighbourhood satisfaction and, in statistical models, renders the objective neighbourhood characteristics insignificant. According to Permentier, this implies that the relationship between neighbourhoods’ objective characteristics and neighbourhood satisfaction is indirect [29].

A neighbourhood’s social environment consists of fellow residents. Social interaction among the residents of the same microneighbourhood is more frequent [30]. This is in line with the notion that neighbourhoods are usually perceived as rather small. Neighbourhood satisfaction is associated with the satisfaction of contact with neighbours [29] (p. 979).

Associated with neighbourhood satisfaction, we also apply socioeconomic factors, such as age, gender, income, and education [23,30], to our analyses. For example, neighbourhood satisfaction has a positive association with income at the household level [30]. In the context of this study, the association between age and neighbourhood satisfaction is crucial, as those most satisfied with their neighbourhoods are the older residents, e.g., [22,28,31]. This association has been elaborated on already by Gory and colleagues [25], who indicated that physical and cognitive decline decreases older peoples’ “action space” and renders them more dependent on environmental conditions. These changes increase the environment’s role as a predictor of action. Moreover, they pointed out that older adults’ lifestyles may differ from those of younger people and, therefore, promote different expectations regarding environments [25].

In addition to neighbourhood satisfaction, including pleasurable and desirable outcomes as such, it also has some positive byproducts. Health and neighbourhood are linked, especially among older adults [26]. Residents’ health is linked to neighbourhood features, such as aesthetics, safety, and walkability [32]. Improving neighbourhood quality can affect neighbourhood satisfaction which, in turn, is an antecedent of various positive outcomes. A neighbourhood can also be seen as a context for networks that constitute a neighbourhood community.

### 1.3. Neighbour Relations and Community

Social ties in the neighbourhood are important sources of wellbeing for its residents. However, neighbourly relations differ notably from our usual social networks; they have been described, for instance, with terms like “acquaintance” or “weak ties” [33,34]. As neighbours, we live close to each other, yet it is possible that we do not even know each other by name. It is also common that the paths of some neighbours do not cross with others due to differences in lifestyles, daily schedules, etc. However, when encounters happen, new resources may be created. According to Völker et al. [35] (pp. 100, 101), we can ‘speak of a community if individuals realise multiple wellbeing goals within the same group of others’—even when the community consists of just a few neighbours. Here, wellbeing goals may refer to both resource-based ones and subjective wellbeing. For instance, neighbour relationships enable various kinds of help from assisting in the laundry room, taking care of a neighbour’s house plants, or giving advice when needed. These are examples of weak ties [34] that usually typify neighbour networks.

Clearly, a sense of community is not given and there can be great variance between neighbourhoods and individuals within them regarding their degree of community. Thus, Völker, et al. [35] define four conditions for neighbourhood community: (a) possibilities to meet (places, synchronised time use), (b) willingness to invest in neighbour relations (accelerated by homophily), (c) not too many alternative competing networks, and (d) reciprocal relationships between residents (interdependency). For instance, spending a notable amount of time in the neighbourhood, being more dependent on others and/or having limited competing social networks are conditions that often apply to older age groups, whereas younger people may have difficulties in devoting their temporal resources to neighbourhood community due to hours spent in work or with family. Moreover, younger residents may have more opportunities to choose their local or virtual networks and those are not necessarily located in one specific neighbourhood.

Concurrently, neighbouring can also be seen as an evolving process. According to Grannis [36], neighbouring evolves when geography allows individuals to be conveniently available to one another, when individuals have passive contacts or unintentional encounters, when they actually initiate contact, and, lastly, when individuals engage in activities indicating trust or shared norms and values. All in all, neighbour networks are still important parts of personal networks despite the growing number of competing alternatives, such as increased mobility and ICT. Moreover, as previous research has shown, neighbour networks have special relevance, especially for older adults; see e.g., [7,37,38]. This paper adds to existing research knowledge by examining older adults’ experiences related to the benefits of age-diverse neighbourhoods and their associations with community wellbeing.

## 2. Data and Methods

The study is based on a mixed-method design [39] and uses both interview and survey datasets. The quantitative data in this study comprises two separate datasets: survey data and a descriptive dataset of condominiums. These datasets were merged based on the street address.

### 2.1. Survey Data 

The survey data were collected via an online questionnaire during the first phase in September 2020. An SMS invitation was sent to all residents who had provided their phone numbers to a rental house company in Eastern Finland (N = 4966). In total, the questionnaire was completed by 420 residents from 115 different condominiums. The questionnaire consisted of thirty-seven questions, most of which were Likert-scale items. Condominium-specific data were acquired through a rental house company. The data includes residents’ ages, genders, and street addresses. From this data, for each of the condominiums, we calculated the mean and the standard deviation of age. The mean age in the condominiums ranged from 18 to 72 years, with an average of 36.8 years. The standard deviation of ages in condominiums ranged from 8.4 to 31.9, with an average of 20.2. On the individual level, the respondents’ ages ranged from 15 to 77 years, with an average of 42.9 years. Most of the respondents were employed (40%), retired (22%), or unemployed (17%), and just over 80% had at least upper secondary education.

In the questionnaire, the respondents also had the option to participate in follow-up studies. Phone-interview participants were recruited based on this consent. The interview data were used to generate hypotheses, which, in turn, were tested with the survey data in the next phase. In the last phase, the interviews were used to investigate what kind of meanings older people give to age in an aged mixed neighbourhood.

The dependent variable was adopted from the Community Wellbeing Index (CWI), which measures an individual’s satisfaction with their local place of residence [24] (p. 733). The CWI comprises three subscales: satisfaction with (1) community services, (2) community attachment, and (3) physical and social environment. As we were interested in residents’ interaction in the neighbourhood, we used only the latter two subscales. The items were rated on a Likert scale, ranging from one to eleven. Regarding reliability, Cronbach’s alphas were 0.85 for ‘community attachment’ and 0.77 for ‘physical and social environment’. Items’ descriptive statistics are presented in Table 1. More details on all of the variables can be found in Appendix A.

People living in condominiums form a natural hierarchy. To take account of this hierarchy, we used hierarchical linear regression. In the following models, condominiums serve as Level 2 observations, and individuals Level 1 observations.

### 2.2. Interview Data

Interview data were collected through semistructured individual interviews after the survey data were collected. The researcher contacted all the online questionnaire respondents aged 65 and over (n = 42) who had expressed interest in participating in the follow-up studies and left their phone number. Out of 42 individuals, 14 were not reached (the phone number was not in use or no one answered the phone) and 6 declined to participate. Three individuals cancelled their participation before their interview was carried out. Hence, the final data consisted of interviews with 19 older residents, aged 65 to 73 years old.

Due to the COVID-19 pandemic, the interviews were carried out by telephone. This was for safety reasons, but also due to travel restrictions in place at the time of data collection (October 2020). The interviewees’ housing histories in their current residences varied from one week to over thirty years and the interviewees came from 14 different districts/neighbourhoods around the municipality. Sixteen of the interviewed individuals were women and three were men. Nine lived alone, six with a spouse or a partner, three with their adult child, and one with an older sibling.

The interviews were semistructured and the themes included questions related to the respondents’ tenement house and neighbours, housing environment and neighbourhood, neighbour relations, and daily life. The researcher who conducted the interviews aimed at giving room for the interviewees to talk about these themes freely and also included in the discussion topics brought up by the interviewees. The interviews lasted between 25 to 80 min; they were audio-recorded with the permission of the interviewee, and transcribed verbatim by a professional transcription company. The transcribed data were analysed by using qualitative content analysis [40]. In the first phase of the analysis, several readings of the transcribed data were conducted. During this phase, units of meaning relevant to the study were identified, i.e., sentences and paragraphs that described relations, views, and experiences with neighbours of different ages. In the next phase, units of meaning were labelled with codes and sorted into categories to discover possible patterns in the data. In the final phase of the analysis, four analysis themes were formed: (1) general attitudes towards residents’ age diversity, (2) social interaction with different aged neighbours, (3) benefits of age-diverse neighbourhoods, and (4) older residents as community builders.

In the following, we will present the results of the study. The quotes shown in the Section 3 have been pseudonymised to ensure the anonymity of the participants.

## 3. Results 

### 3.1. Qualitative Findings

#### 3.1.1. General Attitudes towards Residents’ Age Diversity

The analysis of the interview data revealed a pattern regarding the older residents’ preference for age diversity among neighbours. Some earlier studies too have indicated that older people may prefer to live in age-diverse neighbourhoods [41]. Overall, our data conveyed that good neighbour relations were valued and several individuals noted that, upon retirement or living alone, the significance of neighbours had increased. Our interviewees also mostly described their neighbour relations in a positive light. This notion resonates with earlier studies conveying that attitudes towards neighbours tend to be more favourable among older people [42]. Living in an age-diverse neighbourhood was described as adding an enriching element to one’s living environment and helping to maintain intergenerational relations:

*I think that if there are people of different ages, it is richness. It maintains the interaction between people of different ages*.(A.12.K)

*Young people should see older people and older people should see young people’s life, for life should not be too narrow. I reckon it should be diverse (like this)*.(A.6.K)

According to previous research, intergenerational relations are important since they support positive ageing in place [43] and contribute to the development of generational understanding which is viewed to be vital to the social, economic, and ecological sustainability of contemporary societies [44]. Having both young/er and older people in one’s place of residence can also support the psychosocial wellbeing of the individual as suggested by the comment *one stays sprightlier when it is like this* (A.18.K)

When the interviewees reflected on possible benefits and disadvantages of living in a place where age diversity was great, some did acknowledge that housing consisting of similar-aged peers might be quieter and perhaps result in more communal action. However, at this point in their lives, the interviewees preferred living in a more age-diverse environment and the idea of moving into a senior house, for instance, was rejected as an undesirable option (A.12.K). Moreover, a house where all the residents were old/er, was likened to a nursing home (A.1.K). This notion resonates with a Dutch study according to which older people who were in better physical condition preferred to surround themselves with younger people [20] (p. 1784).

#### 3.1.2. Social Interaction with Different Aged Neighbours

Neighbourhood can be an important context and facilitator for relationships throughout a person’s life course and particularly in later life. Neighbours can provide a significant source of social contact, especially for those living alone and for individuals with decreased physical mobility. In our data, the most common type of daily interaction with neighbours was greeting and exchanging a few words. For some, this was enough since keeping a friendly distance is often preferred in neighbour relations [38]. However, there were also individuals who had formed closer ties with their neighbours. In these cases, the neighbour interaction was usually more frequent and diverse, such as going for walks together, staying connected through social media, giving emotional support, and/or practical help when needed, and even providing care.

Visiting each other’s homes was rare and interaction among neighbours mainly took place in the yard or other common areas of the residential building. Occasionally, events or a work party organised by the house committee (if the residential building had one) provided further opportunities to socialise with the neighbours. Correspondingly, a lack of places to meet, negative associations related to one’s neighbourhood, or simply not having any interest in knowing one’s neighbours were given as the main reasons for minimal neighbour contacts.

Overall, our interviewees socialised more with neighbours who belonged roughly to the same age group as they did. This was partly because working-age neighbours and students were usually away during the daytime when older residents were out and about. As a result, encounters with other “daytime residents” were more likely to occur. It was recognised that differences in daily schedules and lifestyles, combined with the fast turnover of young tenants, made it challenging to get to know the younger neighbours in the housing complex.

However, one’s neighbourhood can offer an important arena for cross-age interaction. When the opportunity to chat with a younger neighbour came, it was often described as refreshing and nice. Also, just seeing younger people in the area and the yard contributed to a sense of a lively neighbourhood where *not all are old grannies* (A.14.K). However, some interviewees felt that it was easier to engage in a “proper” conversation and form friendships with neighbours who were their age peers. This was not necessarily always a consequence of the (assumed) homophily principle in social interaction, but rather a consequence of having lived long in the same house and/or neighbourhood, as the following indicates:

*The fact that we’ve lived here for years…It feels like they are acquaintances, when you constantly see them for many years, in the yard so, it’s easy then and, for that reason too, there is more to talk about*.(A.1.K)

In some cases, chronological age was also seen as less relevant if the neighbour was congenial or had similar interests. For instance, one female interviewee (A.3.K) told how the dog owners of her housing complex often got together in the yard to chat and spend time with each other. Here, a shared interest in dogs and being a dog owner was considered more relevant than belonging to the same age group.

#### 3.1.3. Benefits of Age-Diverse Neighborhoods

When asked about the interviewees’ perceptions about living in an age-mixed neighbourhood, many noted that children in particular made the house premises livelier in a positive way. Even trivial things, such as seeing children running and playing in the yard, hearing “the sound of the little feet” from the adjoining apartment, or children waving “hi” were seen as bringing joy. For people with no grandchildren or whose grandchildren had already grown up, age-mixed housing created opportunities to meet and interact with children, which some of the interviewees greatly appreciated. One woman, whose housing had a daycare centre on the ground floor, talked at length about the topic. From her balcony, she often watched the children putter around in the yard and depicted how lovely it was to be noticed by them:

*You see always, every time I go to the yard […] they notice me right away, that I’m here and so. And then I start talking to them and so forth, and I think it is just so lovely when they always come to me. And then each time I have to tell them where I’m going and when I’m coming back and things like this*.(A.7.K)

Living in a house consisting of differently aged residents was seen as beneficial to both younger and older people. Concurrently, as children brought joy to older residents and invigorated the neighbourhood in general, some interviewees felt that their presence in the neighbourhood contributed to a safe and harmonic environment, as the following extracts indicate:

*Perhaps that too brings them [children] a kind of security when there’s an older person (like me) living in the same yard area too*.(A.4.K)

*Like in the old times when I grew up, there were three generations sitting around the same table. It kind of brings people a little closer. Like, these generation gaps. [Interviewer:] …you mean when there are people from different ages and generations living in the same house? [Interviewee]: Yes. I think it often creates a peaceful atmosphere and perhaps does other good things too*.(A.10.K)

Earlier research has shown that intergenerational and reciprocal roles can become more important with age [45]. At the same time, finding ‘naturally occurring opportunities’ [46] (p. 1785) for intergenerational interaction has become more challenging since, in many European societies, Finland included, age segregation is prevalent and cuts through many domains of society; see [47,48]. However, age-mixed housing can provide a central everyday arena for fostering intergenerational interaction as spatial proximity is crucial for ‘face-to-face interaction to occur and bonds to be built’ [47] (p. 351). For older people with mobility or health issues, the opportunity to meet children and younger people in their immediate living environment can be particularly important—especially for the oldest old whose social life is usually bound to the place of their residence; see [49].

Living in age-mixed housing was also seen as beneficial in terms of the support and help that neighbours could offer to each other. Younger people were considered a good source of support for the aged. Many interviewees described how, during the COVID-19 lockdown in particular, younger neighbours had helped the older residents by delivering them groceries and other necessities.

In addition to receiving concrete help, some interviewees also pointed out that having younger neighbours helped them to keep up with the times, figuratively speaking. One female interviewee, for instance, noted that although she did not interact with her younger neighbours daily, she still found their lifestyles inspirational (A.1.K). Some interviewees also mentioned that chatting with younger neighbours made them feel younger:

*There’s some younger people here who gladly stay for a chat and always then I notice that I’m not that old after all*.(A.18.K)

Correspondingly, the most common way older people helped their younger neighbours was by offering them emotional support, encouragement, and advice. Having plenty of experience from many walks of life was seen as a strength that older individuals could draw from when providing guidance:

*Many older people have lots of knowledge. So, you could, if you’re talking with the youngsters, share the worldly wisdom about life*.(A.11.K)

*Although they say that granny doesn’t know anything, someone can say this, and I’ve heard it too, not here but on a general level, but the granny can know a lot of things that would benefit a younger [person]*.(A.14.K)

Being able to take a role as a mentor can have many positive impacts for older adults. For instance, it can give a sense of being important and needed, which contributes to wellbeing in later life. In addition, having meaningful relationships and roles in one’s neighbourhood can compensate for the negative aspects of one’s living environment. This came up in an interview with a woman who had considered moving to another location. However, her role as a confidante and a support person for some younger neighbours helped the interviewee to cope with the experienced downsides of the living environment:

*I playfully always call our bunch ‘our youth’ [laughs] since I’m the oldest one in this house. One of them actually said nicely that [I am] ‘the mamma of our house’ (…) And sometimes I’ve had this thought of moving out from here because that street traffic bothers me, but these three people said to me: ‘Don’t move anywhere, don’t go’. It felt really good*.(A.13.K)

#### 3.1.4. Older Residents as Community Builders

Building a sense of community can be challenging in a neighbourhood characterised by a relatively fast turnover of residents, less interaction with neighbours, and little communal activity, as the interviews also conveyed. For some, the lack of community was not an issue if their social life took place mostly outside of the neighbourhood or if they preferred spending time alone. Some interviewees, however, longed for more neighbour interaction and communality:

*Of course, a community where you do things together, or something, would be nice. And, if you needed help, you wouldn’t have to be shy to ask for it. So, like that, if we knew each other better, if we were more interested in each other…But now, everyone is sort of minding their own business, and thinking that they can take care of their things by themselves*.(A.K.17)

Some qualitative studies have indicated that older residents can have a meaningful role in generating a sense of community and connectedness in the neighbourhood [50,51], which also speaks for the significance of age mix in the neighbourhood. In our study, many of the interviewees aimed at affirmative interaction with the neighbours by greeting everyone who lived in the same building, by welcoming new residents when possible, and by conducting varying acts of kindness for the neighbours. They were also always up for chatting with neighbours. These can all be seen as contributing to a neighbourly atmosphere and potentially building a sense of community and connectedness among the residents. Some also pointed out that they tried not to be judgmental about other people and how they lived their lives, as the following excerpt suggests:

*Sometimes the young people might cause some problems and the police might show up and the like, but the way I see it, it’s none of my business. I take them [the young people] as human beings as I do with everyone too*.(A.11.K)

Given the importance of neighbourhood to many older adults, the local community can be significant to the individuals’ experience of social exclusion but, concurrently, also to the experience of inclusion [52]; see also [18]. Friendly encounters among neighbours can greatly contribute to a sense of being included in the community, as the following excerpt also suggests:


*There’s this couple with two kids downstairs, with them I chat more. They are nice people, and they talk to me as an equal and they don’t think, what are those old people doing here, shouldn’t they be living somewhere else?*
(A.7.K)

### 3.2. Quantitative Findings

In Table 2 are presented the results from the regression analyses. Interestingly, the findings from our survey data suggest that with age, satisfaction towards community attachment slightly decreases (Table 2). Concurrently, as age is seen to bring an increasing attachment to one’s neighbourhood, it can also bring an increased vulnerability and sensitivity to one’s physical and social environment [53] (p. 592). One possible explanation for this finding is that the immediate neighbourhood does not meet the expectations and needs of the oldest residents when it comes to the social environment. Given that the social life of ‘the oldest of the old’ is usually bound to their place of residence [49], and that older people can be more dependent on their neighbours for support [54] (p. 10), the lack of interaction with neighbours can negatively affect the experience of social wellbeing and, thus, also decrease the level of satisfaction towards community attachment.

Based on the qualitative findings, we may hypothesize that age diversity is associated with community attachment and satisfaction with the neighbourhood’s environment. To test this, we made two similar sets of analyses for the two dependent variables, community attachment (CA) and physical and social environment (PS). The first model includes background variables, such as gender, employment status, family relations, and income. As familiarity with neighbourhood and attachment to place arguably grows over time, we decided on controlling the number of years dwelled at a current location. The mean and standard deviations of age in condominiums were also included.

Other themes which arose from the interviews were examined in the second and third models. The “social interaction” is examined in the second model by adding two variables that measure social interaction with neighbours. Both had a positive association with CA but not with PS. The less an individual interacts with neighbours, in the form of chatting or otherwise spending time with them, the less they are satisfied with CA. In the third model, we analysed the effects of perceived disturbances in the neighbourhood. The results show that the more an individual thinks children cause disruption in the neighbourhood, the more dissatisfied he/she is with PS.

Overall, satisfaction with the neighbourhood seems to decline with age. The direct association of age with CA was slightly negative; for PS, the association was even stronger. The standard deviation of age also had a negative effect on PS. However, age and standard deviation of age in the condominium had an interaction effect. According to our data, satisfaction with CA and PS declines with age, if the age structure of the condominium is homogenous. If the age structure is heterogenous enough, ageing had a positive association with CA and PS. These interaction effects are depicted in Figure 1 and Figure 2.

## 4. Discussion

The results of our study indicate that older adults in particular seem to benefit from age diversity in the neighbourhood; they reported higher community wellbeing and considered age diversity as advantageous for them. In the interviews, living in an age-diverse neighbourhood was described as rich and positive encounters with young(er) neighbours evoked feelings of joy and inclusion among the older residents. Age diversity among the residents offered opportunities to meet age others and even take intergenerational and reciprocal roles in terms of providing companionship, emotional support, and practical help to one another. Some of our interviewees felt that their mere presence contributed to a safer atmosphere in the neighbourhood, which calls attention to the diverse ways that individuals can function as a resource in the community.

For older persons, age diversity in one’s living environment may provide opportunities to interact with the younger generation, to provide guidance, and to “give back”, i.e., participate in generative activities, which is known to have a positive impact on quality of life [55,56]. As Völker et al. [35] argued, there are certain conditions under which neighbourhoods or communities can produce wellbeing outcomes; for example, spending more time in the neighbourhood and having limited competing social networks are conditions that often apply to the older adult population, whereas younger residents’ main social networks and activities are not necessarily located in the neighbourhood.

The interviewees also spoke about the importance of seeing and meeting individuals of different ages in one’s living environment as a way to maintain intergenerational relations and reduce negative cultural perceptions of age others; cf. [57]. This is no minor issue, considering that critical gerontology has provided plenty of research evidence on negative stereotypes, attitudes, and prejudices towards older people, which does not exactly help in bridging possible generational gaps or building intergenerational solidarity. Some of the interviewees also talked about ageist prejudices younger people may have against older people. However, seeing and interacting with neighbours of different ages can enable the development of generational understanding and ‘the ability to put oneself in the position of other generations’ which is necessary for achieving more harmonious intergenerational relations [44] (p. 118).

The social fabric of the neighbourhood and a sense of being included in one’s community are important components of wellbeing in older age. While our results convey older adults’ preference for age diversity in one’s living environment, it is important to remember that older people are a very heterogenous group with different life situations, needs, and housing preferences. For instance, some studies suggest that people over seventy-five may benefit from age-segregated housing; see [58], cf. [59]. A few of our interviewees pondered that their housing preferences in terms of age diversity might change as they age. Young neighbours may also arouse mixed feelings among some older adults and age diversity can potentially also have a negative effect on neighbourhood attachment; see [1].

The study has limitations that are important to consider. The age diversity of the interview participants was relatively narrow and failed to capture the views and experiences of persons aged 74 and older. This was likely because the interview participants were recruited from among the residents who had answered the online questionnaire. Existing research, however, shows that online activity and the use of digital technology decline for various reasons as people grow older; see [60]. Consequently, to reach representatives of older age groups in future studies, more traditional recruiting methods should be used. In addition, the interviewees were mostly women and, for instance, ethnic minorities were not represented in the study. In this regard, the study does not provide possibilities for generalizing the results to the wider population of older people.

In the quantitative data, the number of observations in multilevel models was small on an individual level. Due to the grouping of individuals, the standard errors and statistical tests are conservative towards the null hypothesis. However, we were more interested in the Level 2 results. On the condominium level, the number of observations was large enough (n = 115) for multilevel analysis. Nevertheless, we checked whether the number of observations in a condominium had an effect on the results. To do this, we removed one by one the condominiums with only one, two, three, or four respondents and ran the analyses. The results did not change.

## 5. Conclusions

This article focused on investigating older residents’ perceptions of age-diverse neighbourhoods and the possible impact of age diversity on community wellbeing. Our results show that older adults may prefer age diversity in their living environment and, simultaneously, also benefit from the age diversity of residents living close by; the larger the variance in age within the neighbourhood, the higher they reported community wellbeing. However, we did not find such an association among the younger residents, which suggests that the impact of age diversity may not be the same for all people and that the importance of certain neighbourhood characteristics fluctuates over a person’s life course.

Our study produced new knowledge on the interplay between age diversity and community wellbeing in a context where intergenerational encounters occur naturally. Our findings also contribute to the discussions on how to create neighbourhoods that correspond better to the preferences and needs of older adults, especially in terms of their social dimensions. The results of this study can be utilized when developing age-friendly environments and housing policies both at local and national levels. Promoting age diversity in neighbourhoods can help communities and cities become more “age-friendly”, which supports healthy ageing; see [61], and which as a policy is seen as central in adjusting to the challenges of population ageing and urban growth [62] (p. 100).

However, future research is needed to investigate the optimal balance or preferred ratio of age variation among residents to enhance neighbourhood satisfaction. Also, interviewing young(er) residents and exploring their views is important in gaining a deeper understanding of the potential benefits of age diversity in a neighbourhood.

## Figures and Tables

**Figure 1 ijerph-20-06574-f001:**
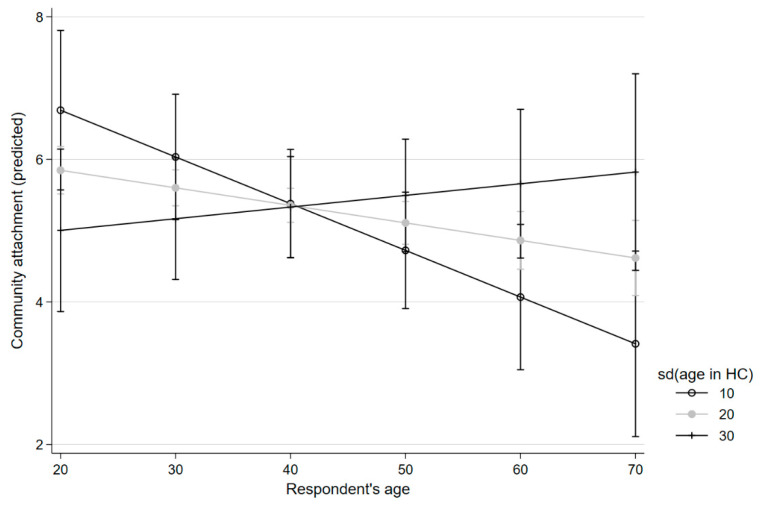
Age, community attachment, and deviation in age in the housing community.

**Figure 2 ijerph-20-06574-f002:**
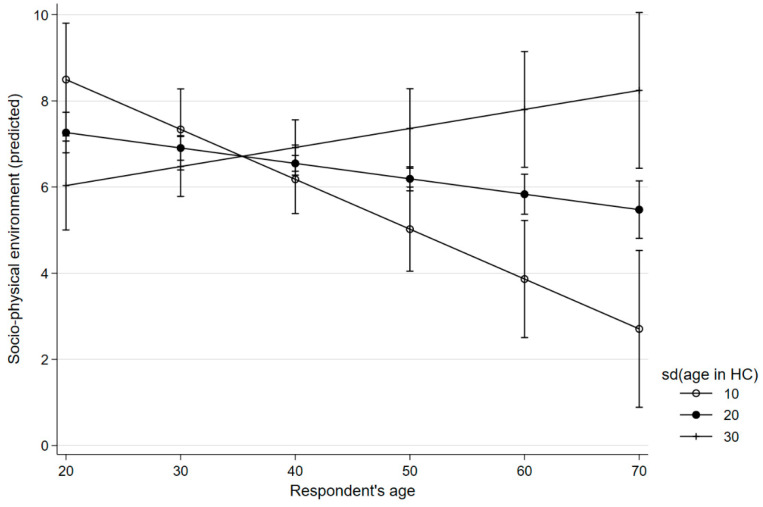
Age, satisfaction with the neighbourhood, and deviation in age in the housing community.

**Table 1 ijerph-20-06574-t001:** Descriptive statistics of the dependent variables.

	n	Mean	Std.Dev.	Min	Max
Community attachment					
belonging	419	6.06	2.44	1	11
trust in people	419	6.11	2.53	1	11
safety	419	7.02	2.67	1	11
Physical and social environment					
environment	419	7.93	2.44	1	11
social conditions/problems	418	5.72	2.39	1	11
economic situation	419	6.16	2.14	1	11

**Table 2 ijerph-20-06574-t002:** Predictors of satisfaction with community attachment and the physical and social environment (hierarchical linear regression).

	Community Attachment						Physical and Social Environment					
**Gender (ref. Woman)**	−0.051		−0.11		−0.072		0.085		0.054		0.101	
**Work status (ref. Employed)**												
*Unemployed*	0.251		0.239		0.324		−0.092		−0.095		−0.181	
*Student*	0.488		0.453		0.399		0.092		0.077		0.043	
*Disability pension*	−0.333		−0.331		−0.301		−0.023		−0.027		0.063	
*Retired*	0.29		0.393		0.349		**0.572**	*****	**0.615**	*****	0.479	
*Housewife or househusband*	0.816		0.461		0.659		0.232		0.082		0.223	
*Other*	0.999		0.815		0.797		0.774		0.703		0.645	
**Kids (ref. no kids)**												
*No kids*	−0.108		−0.153		−0.067		0.122		0.108		0.231	
**Relationship (ref. in relationship)**												
*Not in relationship*	−0.228		−0.142		−0.144		0.017		0.05		0.268	
**Income**	0.07		0.086		0.063		**0.126**	*****	**0.133**	******	**0.149**	******
**Years dwelled**	0.008		0.005		0.006		0.007		0.006		0.012	
**Loneliness**	−0.15		−0.104		−0.129		**−0.248**	*****	−0.228		**−0.284**	*
**Age**	**−0.106**	******	**−0.108**	******	**−0.107**	******	**−0.209**	*******	**−0.209**	*******	**−0.196**	*******
**Mean age of condominium**	−0.017		−0.015		−0.015		−0.017		−0.017		−0.011	
**sd(age of condominium)**	−0.182		−0.172		−0.166		**−0.335**	*******	**−0.328**	*******	**−0.283**	******
**age * sd**	**0.004**	*****	**0.004**	*****	**0.004**	*****	**0.009**	*******	**0.009**	*******	**0.008**	*******
**How often do you spend time with your neighbours**			**−0.101**	*****	**−0.094**	*****			−0.046		−0.088	
**How often do you chat with your neighbours**			**−0.119**	*****	**−0.144**	*****			−0.044		−0.016	
**Different-aged children and young people cause disruption in the neighbourhood and the neighbours just have to accept it as a part of everyday life.**					−0.0486						**−0.251**	**
**Observations**	306		306		296		306		306		296	
**var(condominium)**	0.039		0.067		0.071		0.000		0.000		0.000	
**var(individual)**	2.207		2.117		2.151		2.790		2.778		2.698	
**AIC**	1013.8		1009.2		958.1		1072.5		1075.3		1010.6	
**BIC**	1084.6		1087.4		1035.6		1143.2		1153.5		1091.8	
**chi2**	119.1		4205.6		892.2		907.6		2201.7		8159.3	
**df_m**	16		18		19		16		18		19	

Unstandardized coefficients. * = *p* < 0.05; ** = *p* < 0.01; *** = *p* < 0.001. Statistically significant estimates are bolded.

## Data Availability

The data presented in this study are available on request from the corresponding author. The data are in the process of being archived and are not publicly available yet.

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
