# Peer review of "Age Diversity in Neighborhoods—A Mixed-Methods Approach Examining Older Residents and Community Wellbeing"

_ijerph, 2023, doi:10.3390/ijerph20166574_

Round 1
Reviewer 1 Report
Overall, this is an interesting manuscript and provides good information about the importance of naturally-occurring opportunities for intergeneration interaction and relationships.
1. Please describe the specific methods and procedures used for the content analysis in the Methods section.
2. Please describe the results of the content analysis in a more systematic way in the Results section.
3. Please provide an overview of the neighborhood and/or condominium context. It would be helpful to know more about the age ranges of residents, the socioeconomic factors, neighborhood safety, community resources, etc. in order to better understand the results.
Author Response
Please describe the specific methods and procedures used for the content analysis in the Methods section. Please describe the results of the content analysis in a more systematic way in the Results section.
Thank you for this comment. In the revised paper we have described the process of the content analysis and the results of the analysis in more detail. We added the following paragraph to the Data and Methods section:
The transcribed data were analysed by using qualitative content analysis (Kleinheksel et al. 2020). In the first phase of the analysis, several readings of the transcribed data were conducted. During this phase, units of meaning relevant to the study were identified, i.e., sentences and paragraphs that described relations, views and experiences with neighbours of different ages. In the next phase, units of meaning were labeled with codes, and sorted into categories to discover possible patterns in the data. In the final phase of the analysis, four analysis themes for formed: 1) General attitudes towards residents age-diversity, 2) Social interaction with different aged neighbours, 3) Benefits of age-diverse neighborhoods and 4) Older residents as community builders.
Please provide an overview of the neighborhood and/or condominium context. It would be helpful to know more about the age ranges of residents, the socioeconomic factors, neighborhood safety, community resources, etc. in order to better understand the results.
We added descriptive statistics and some details to the ‘Data and methods’ section. Unfortunately, we do not have data on neighbourhood safety, community resources, etc. available at the moment.
Reviewer 2 Report
Thank you for the opportunity to review this article. I offer this feedback in the spirit of collegial support and I do hope it is helpful.
This mixed-method study set out to answer 3 important questions, all integral to understanding the value of age diverse communities for community well-being. Do age-diverse communities contribute to community wellbeing? If so, ‘who benefits’ and ‘how’?. The study potentially contributes to the international literature on creating supportive neighbourhoods for older adults (aged 65-75 years). The authors establish this is an under-researched area in Finland and therefore will have national relevance.
The article is well organised and contains all components expected of research article. Overall, the literature is well synthesised for factors influencing neighbourhood and community wellbeing. It would be helpful to have a definition of neighbourhood and community wellbeing so the parameters for this study are clear for the reader. It is disappointing to have so many old references when there is a good amount of more recent age-friendly research that would be relevant for this study. Many of the references are very old and in this instance [ 168/169] it makes one question the relevance of the data. I recommend including more recent literature.
All sections are well covered with the exception of Data and Methods. The qualitative data is extensively reported but the quantitative data is insufficiently reported. In results 435-449 the authors need to provide more explanation of the data to help the reader make sense of the tables. In other words help the reader navigate round the tables and eg provide some statistical interpretation of the significance of the association between factors. I assume the asterix indicates some significance but this needs to be explained. 206-232 would benefit from tighter editing and greater clarity It appears hierarchical linear regression was used to anlayse a very small data seton such a small number needs to be identified as a limitation.
Has the study answered the questions? Not clearly. The data included diverse ages yet the data is analysed from the perspective of the older residents only. Therefore it can’t answer the questions ‘does age diversity contribute to community well-being? Or ‘ mechanisms explaining the association between the residents’ age diversity and well-being of the community. Perhaps the questions would benefit from a review in relation to the findings and discussion.
There needs to be a conclusion to clearly identify what new knowledge this study contributes to the international literature and how these findings could possibly be used and by whom.
Thank you for giving some thought to this feedback.
A section on limitations of the study needs to also be included.
The title needs to better reflect the study. Contrary to the title the research has nothing to do with housing for age-diversity. While the research is conducted with residents of condominium style accommodation the study is interested in satisfaction with age-diverse neighbourhoods and not housing.
The abstract needs to better reflect the intention to explore the perspective of older residents. It is evident early on (line 33) that the research will be from the perspective of older residents of the neighbourhood but this is not evident in the abstract.
The researchers need to address limitations of the study
Thank you
Author Response
The title needs to better reflect the study. Contrary to the title the research has nothing to do with housing for age-diversity. While the research is conducted with residents of condominium style accommodation the study is interested in satisfaction with age-diverse neighbourhoods and not housing. (…) The abstract also needs to better reflect the intention to explore the perspective of older residents.
We have revised the title of the article and the abstract to clarify that the emphasis of our study is on older residents. We hope the new title, “Age-diversity of neighbourhoods – a mixed methods approach examining older residents and community wellbeing” reflects the study better. Age diversity, as such, is a central theme of our study which is approached with both interview and survey data. We thank the reviewer for the useful remarks regarding neighbourhoods and neighbouring instead of housing.
It would be helpful to have a definition of neighbourhood and community wellbeing so the parameters for this study are clear for the reader.
We added a definition of community wellbeing on page 2, under the Introduction chapter. Neighbourhood satisfaction and community well-being are used synonymously here, although a very careful examination might identify differences between them.
The quantitative data is insufficiently reported. In results 435-449 the authors need to provide more explanation of the data to help the reader make sense of the tables. In other words help the reader navigate round the tables and eg provide some statistical interpretation of the significance of the association between factors. I assume the asterix indicates some significance, but this needs to be explained. 206-232 would benefit from tighter editing and greater clarity. It appears hierarchical linear regression was used to analyse a very small data seton such a small number needs to be identified as a limitation.
As suggested, the section that describes the interview data collection is more compact in this latest version of the manuscript. Regarding quantitative data, we have clarified the results section. Explanations for the asterisks were added on the tables and the caption of Table 2 was revised. A description of variables can be found in Appendix 1. The number of observations is discussed in the limitations of the study section (see ‘Discussion’). The interpretation of the tables and figures is now written in a more detailed manner.
It is disappointing to have so many old references when there is a good amount of more recent age-friendly research that would be relevant for this study. Many of the references are very old and in this instance [168/169] it makes one question the relevance of the data. I recommend including more recent literature.
Thank you for this comment. As suggested, we have revised the references and included more recent literature in this new version of the manuscript.
Has the study answered the questions? Not clearly. The data included diverse ages yet the data is analysed from the perspective of the older residents only. Therefore it can’t answer the questions ‘does age diversity contribute to community well-being? Or ‘ mechanisms explaining the association between the residents’ age diversity and well-being of the community. Perhaps the questions would benefit from a review in relation to the findings and discussion.
As recommended by the reviewer, we have reformulated the research questions presented in the ‘Abstract’ and in the ‘Introduction. The revised ones are:
The aims of our paper are: a) to investigate whether age diversity in neighbourhoods contributes to older residents' well-being and b) to explore older residents’ experiences and views on age diversity in their neighbourhood.
The researchers need to address the limitations of the study.
Thank you for pointing this out. We added two paragraphs to the ‘Discussion’ where we address the limitations of our study:
The study has limitations that are important to consider. The age diversity of the interview participants was relatively narrow and failed to capture the views and the experiences of persons aged 74 and older. This was likely because the interview participants were recruited from among the residents who had answered the online questionnaire. Existing research, however, shows that online activity and the use of digital technology declines for various reasons as people grow older (see e.g., Gallistl et al. 2020). Consequently, to reach representatives of older age groups in future studies, more traditional recruiting methods should be used. In addition, the interviewees were mostly women, and, for instance, ethnic minorities were not represented in the study. In this regard, the study does not provide possibilities for generalizing the results into the wider population of older people.
In the quantitative data the number of observations in multilevel models were small on individual level. Due to the grouping of individuals, the standard errors and statistical tests are conservative towards the null hypothesis. However, we were more interested in the Level-2 results. On the condominium level the number of observations was large enough (n = 115) for multilevel analysis. Nevertheless, we checked whether the number of observations in condominium had an effect on the results. To do this, we removed one by one the condominiums with only 1, 2, 3, or 4 respondents and ran the analyses. The results did not change.
There needs to be a conclusion to clearly identify what new knowledge this study contributes to the international literature and how these findings could possibly be used and by whom.
As suggested by the Reviewer, we have added ‘Conclusions’ to the manuscript.